# The Reliability and Compatibility of the Paper and Electronic Versions of the POLLEK Cohort Study Questionnaire

**DOI:** 10.3390/healthcare8040438

**Published:** 2020-10-29

**Authors:** Maksymilian Gajda, Szymon Szemik, Katarzyna Sedlaczek, Małgorzata Kowalska

**Affiliations:** 1Department of Epidemiology, School of Medicine in Katowice, the Medical University of Silesia, 18 Medyków Street, 40-752 Katowice, Poland; mkowalska@sum.edu.pl; 2Department of Nursing Propaedeutic, School of Health Sciences, the Medical University of Silesia in Katowice, 20/24 Francuska Street, 40-027 Katowice, Poland; sszemik@sum.edu.pl; 3School of Medicine in Zabrze, the Medical University of Silesia, Traugutta 2 Square, 41-800 Zabrze, Poland; katarzyna.sedlaczek@wp.pl

**Keywords:** cohort study, questionnaire validation, quality of life, burnout syndrome, mental health, alcohol addiction, medical students, young physicians, WHOQOL-BREF, AUDIT

## Abstract

*Background*: Chronic fatigue, depression, burnout syndrome, and alcohol addiction have been identified as significant mental health problems in young medical doctors. Given the lack of prospective studies in this area in Poland, the POLski LEKarz (POLLEK) cohort study was created. The goal of the POLLEK study is to assess the quality of life and health status (including mental health) of medical students and young physicians. The aim of the presented paper was to assess the reliability and compatibility of paper and electronic versions of the POLLEK questionnaire. *Methods*: Between 1 October 2019 and 28 February 2020, all medical students (*N* = 638) of the first year in the Medical University of Silesia were invited to participate in a cross-sectional study. Three hundred and fifty-three students (55.3%) who accomplished both versions were included in the current analysis. *Results*: Values of Cronbach’s alpha >0.7 proved both modes of delivery to have good internal consistency, except for the individual Alcohol Use Disorder Identification Test (AUDIT) domains and the Environmental domain of the WHOQOL-BREF (paper version). Similarly, interclass correlation coefficients equal to or greater than 0.9 denoted an excellent reproducibility. *Conclusions*: We documented very good accordance and reproducibility of POLLEK questionnaire (both paper and electronic versions). These findings legitimize the use of the questionnaire interchangeably.

## 1. Introduction

The current epidemiological situation of COVID-19 in all European countries (including Poland) vividly highlighted how difficult and responsible medical doctor work is. A previously published review paper of our team revealed that psychosocial determinants have a significant impact on mental health and quality of life of physicians [1]. Chronic fatigue, burnout syndrome, alcohol addiction, risky alcohol consumption, depression, and potential suicidal ideation are among the most important mental health problems of young medical doctors and even medical students [2,3,4,5]. Public health experts suggest that future research of mentioned problems should be conducted on the bases of prospective observations. The lack of this type of research in Poland justifies taking up the research topic in the group of medical students, as future doctors. We understand that reliable scientific knowledge requires appropriate, standardized tools, including validated research questionnaires. 

Following the mentioned justification, we created the integrated original questionnaire that we used in the first step of POLLEK cohort study, which aimed to identify and evaluate the quality of life and health status (including mental health) of medical students and young physicians with simultaneous assessment of their determinants related to studying and working conditions in medical students and young physicians during a long-term observation. Additionally, in the model of epidemiological cohort study, a control of the socio-demographic factors, as well as those that identify lifestyle and chronic diseases is planned. The aim of the presented paper is an evaluation of the reliability and compatibility of both the paper and electronic versions of the POLLEK questionnaire.

## 2. Materials and Methods

### 2.1. Study Design and Sampling

A cross-sectional study was performed between 1 October 2019 and 28 February 2020. All medical students (*N* = 638) of the first year in the Medical University of Silesia (MUoS, Poland) were invited to participate in the study project. Written consent to the examination was obtained from *n* = 559 students (*N*_1_ = 354; 91.2% of all medical students in Katowice and *N*_2_ = 205; 82.0% of all medical students in Zabrze); both are medical faculties of MUoS. Detailed descriptive statistics were presented in Table 1.

The first step of the study was related to the necessity of questionnaire validation. The integrated tool includes the Polish version of the WHOQOL-BREF questionnaire [6], the next is the Alcohol Use Disorder Identification Test (AUDIT) [7], and also the original questionnaire identifying individual nutrition, demographic, socioeconomic, and anthropometric determinants. It is worth indicating that the WHOQOL-BREF questionnaire regards four domains of quality of life (26 items in total): somatic (physical health), psychological, social (social relationships), and environmental domains. Whereas, the AUDIT questionnaire is a 10-item screening tool to assess alcohol consumption, drinking behaviors, and alcohol-related problems. Both questionnaires had been successfully used in previous studies [8,9].

### 2.2. Statistical Analysis

Initially, data were analyzed using descriptive statistics (median and interquartile range, IQR). Reproducibility (test–retest reliability of the POLLEK questionnaire) was assessed by asking all of the students (*N* = 638 in both medical faculties of MUoS) to complete the paper and online version of the instrument. A total of 560 students (response rate of 87.8%) completed the paper version (341 females, 218 males, and 1 missing data). As many as 353 students (55.3%) also completed the electronic version of the questionnaire; nearly 62% of them were women. The median age of respondents was 19 years. About three-quarters of students were living away from their families. Detailed statistics are presented in Table 1. The interclass correlation coefficient (ICC) was analyzed in a test–retest reliability study using the ICC function available in the psych (v1.9.12) package in R software. Moreover, the Bland–Altman plots were obtained to describe differences between the scores and assess heteroscedasticity [10]. Additionally, the repeatability was evaluated by Cohen’s kappa statistics [11]. The reliability of the scales and their domains was evaluated using Cronbach’s alpha coefficients of internal consistency. Moreover, we conducted confirmatory factorial analysis (CFA) using the lavaan (v0.6–5) package in R software to evaluate the structure of each major part of the questionnaire and their domains. To measure the goodness of fit, the Comparative Fit Index (CFI), Tucker Lewis Index (TLI), and root mean square error of approximation (RMSEA) were used. RMSEA results were scored as a good fit for ≤0.05, adequate fit (0.05–0.08), mediocre (0.08–0.10), while values > 0.10 denoted not acceptable fit. Furthermore, values of CFI and TLI greater than 0.95 were interpreted as an acceptable fit [12,13].

All analyses were performed in R 3.6.2 software [14], and results were presented with the respective confidence intervals (95%) or p values (significant at the level <0.05).

### 2.3. Ethical Approval 

The ethics approval for the study was received from the Bioethical Committee of the Medical University of Silesia in Katowice (approval number KNW/0022/KB/217/19; date: 8 November 2019). Written informed consent was obtained from all participants.

## 3. Results

The Bland–Altman analysis demonstrated high accordance between scores in the paper and internet version of WHOQOL-BREF and AUDIT scales (see Figure 1 and Table 2 for more details).

In the ICC analysis, the accordance between paper and electronic version of the WHOQOL-BREF questionnaire was excellent for the overall scores (ICC = 0.92), and the specific domains (ICCs vary from 0.90 to 0.94). Additionally, assessment of repeatability of answers to particular questions (with Cohen’s kappa, Spearman’s rho, and Kendall’s tau) is available in supplementary Appendix A.

Both versions of the WHOQOL-BREF questionnaire had very good internal consistency (**α** near or equal to 0.9), while the reliability of the electronic version was higher compared to the paper. The greater difference was revealed for the environmental domain. The accordance between both versions of the AUDIT questionnaire was also excellent (ICC value of 0.96), including specific domains except for the “Dependence Symptoms” domain with a value of 0.83. We demonstrated also a good internal consistency (α value of 0.77), except for the individual AUDIT domains. Detailed results are presented in Table 2.

The model fit indicators of CFA-RMSEA of 0.067 (90%CI 0.062–0.072) demonstrated a good model fit of the four-dimensional structure of the Polish WHOQOL-BREF questionnaire, whereas CFI and TLI values of 0.821 and 0.799, respectively, indicated a somewhat poorer fit. For the AUDIT questionnaire, an adequate model fit was shown, with CFA-RMSEA value of 0.072 (90% CI 0.054–0.09), CFI = 0.931, and TLI = 0.903.

## 4. Discussion

Reliability and reproducibility are important aspects of questionnaire validation. Questionnaires should be able to reproduce results to be valid [15]. Regarding the statistical measures using commonly in validation studies, the results of the reviewed bibliography indicated that the questionnaire validity was assessed mainly by Cronbach’s alpha coefficient and intraclass correlation coefficient (ICC) [16,17] This observation showed that these measures were used in our study in a reasonable manner. Moreover, obtained results documented well or very good reproducibility (ICC > 0.8 and Kappa Cohen > 0.8 in each assessed domain). Additionally, the results of the measured Cronbach’s alpha statistic confirmed moderate or high consistency (α > 0.5 in each scale). 

The possibility of using the paper version of WHOQOL-BREF in many populations has its established position [8,18]. Few studies have been conducted using the WHOQOL-BREF questionnaire to assess the impact of medical education on the quality of life of students [19,20,21,22]. Although the role of the electronic version of the WHOQOL-BREF questionnaire was confirmed in 2008 [23,24], we have not been able to find a study assessing the electronic form in the medical student population. To the best of the authors’ knowledge, the presented study is the first study authorizing the use of the online version of this tool among medical students.

The AUDIT is a screening questionnaire developed by the World Health Organization (WHO) to assess alcohol-related problems, and available published data indicate that it is a reliable and valid tool used in different cultural backgrounds [25,26,27,28,29,30]. However, the AUDIT questionnaire has not yet been validated among medical students in Europe, both in the paper or electronic form. Nevertheless, the electronic version of this tool was used in a validation study among medical students from China [31] and was validated among university students [32,33]. It is worth mentioning that the AUDIT questionnaire was applied in a cross-sectional study on the prevalence of alcohol use disorders among American surgeons [34]. We believe that the findings described in the presented paper can complement the observed gap.

Obtained results confirmed, that both versions (paper and electronic) of the AUDIT and WHOQOL-BREF questionnaires can be used interchangeably in the Polish cohort study of medical students and young medical doctors. Very high agreement in both kappa and ICC statistics (higher than 0.8 in each case of the assessed domain) indicates that an electronic questionnaire is a reliable tool in planned cohort studies aimed to assess the quality of mental health. This is an important observation for future planned research that will be realized during the COVID-19 pandemic when real-time interpersonal contacts are significantly hampered. Choosing the electronic version of the tool will facilitate contact with medical students in the coming years, also after completing education, and at the same time, will significantly reduce the costs of subjects’ recruitment. In general, it can be assumed that the results obtained by us are consistent with previous observations [8,20,21,24,25,26,28,29,31,32]. 

## 5. Limitations of the Study

Although a large proportion of the invited students have agreed to participate in the study, the fact of not obtaining one hundred percent participation may to some extent limit the conclusions of the study. Similarly, as Chen et al. reported [23], we have only examined the validity of the Internet version of WHOQOL-BREF questionnaire with a standard set of items, whereas the WHOQOL group recommended adding some questions relevant for the studied population [35]. Although our integrated questionnaire also contained additional demographic and nutritional assessment questions, they were not an extension of the WHOQOL-BREF questionnaire.

## 6. Conclusions

We demonstrated very good accordance and reproducibility of both versions of the POLLEK questionnaire. These findings legitimize the use of both versions interchangeably.

## Figures and Tables

**Figure 1 healthcare-08-00438-f001:**
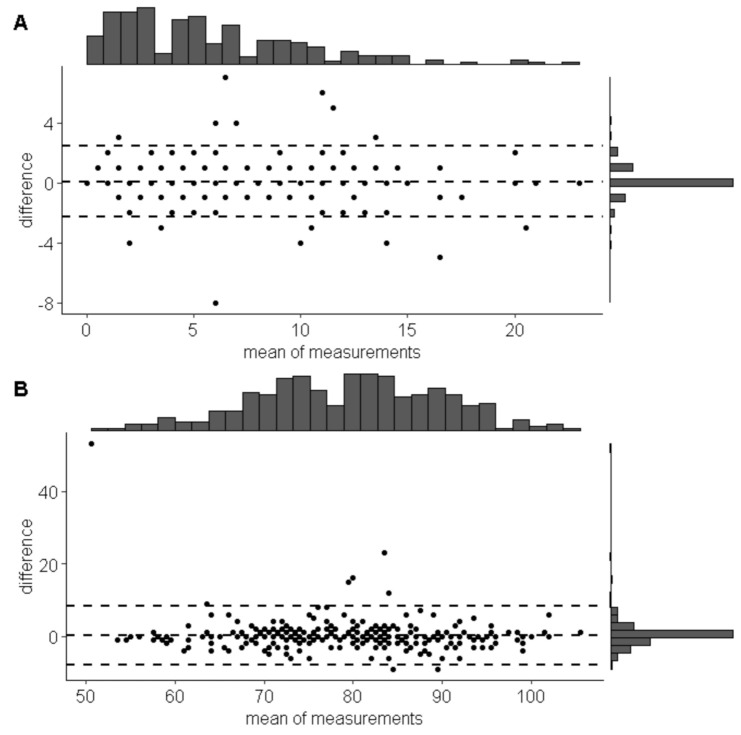
Bland and Altman plots showing the differences between scores (**A**—whole Alcohol Use Disorder Identification Test (AUDIT) scale; **B**—WHOQOL BREF) in paper and internet mode of delivery. The central dashed line denotes the average difference between scores, while the two peripheral lines represent means with ±1.96 standard deviations.

**Table 1 healthcare-08-00438-t001:** Basic descriptive statistics identifying demographic and socio-economic situation of medical students (*N* = 353).

Quantitative Variable—Median and Interquartile Range (IQR)		
M2. Age [years]	Median (IQR)	19.0 (19.0–20.0)
	Missing	1 (0.3%)
Qualitative Variables—*N* (%)		
M1. Gender	Female	218 (61.8%)
	Male	135 (38.2%)
M3. Marital status	Married or domestic partnership	112 (31.7%)
	Single	239 (67.7%)
	Missing	2 (0.6%)
M8A. Source of income—dependent on parents	Yes	337 (95.5%)
	No	16 (4.5%)
M8B. Source of income—scholarship	Yes	20 (5.7%)
	No	332 (94.1%)
	Missing	1 (0.3%)
M8C. Source of income—pension or allowance	Yes	10 (2.8%)
	No	343 (97.2%)
M8D. Source of income—paid work	Yes	37 (10.5%)
	No	316 (89.5%)
M8E. Source of income—other	Yes	15 (4.2%)
	No	337 (95.5%)
	Missing	1 (0.3%)
M9. Current financial situation	Very poor	2 (0.6%)
	Poor	4 (1.1%)
	Average	83 (23.5%)
	Good	150 (42.5%)
	Very good	114 (32.3%)
M10. Current place of residence during studies	Family home	101 (28.6%)
	Dormitory	36 (10.2%)
	Flat/room rented or owned	216 (61.2%)

**Table 2 healthcare-08-00438-t002:** Statistics for scales and their domains for both methods of delivery and measures of reproducibility.

Dependent Variable	A. Descriptive Statistics and Internal Consistency												B. Measures of Reproducibility (Paper vs. Internet)					
	Paper Version						Internet Version						Bland–Altman Statistics		Cohen’s Kappa		ICC	
	Me	IQR	M	SD	α	95% CI	Me	IQR	M	SD	α	95% CI	MoD	95% CI	κ	95% CI	ICC	95% CI
**AUDIT**	5	2–8.25	5.74	4.41	0.77	0.74-0.8	4	2-8	5.55	4.45	0.77	0.73–0.8	0.07	−0.05–0.2	0.88	0.87-0.9	0.96	0.96–0.97
Hazardous Alcohol Use	3	2–5	3.63	2.27	0.67	0.6–0.73	3	2–5	3.52	2.27	0.67	0.6–0.73	0.05	−0.01–0.12	0.87	0.85–0.9	0.96	0.96–0.97
Dependence Symptoms	0	0–1	0.7	1.05	0.51	0.37–0.6	0	0–1	0.69	1.11	0.55	0.44–0.65	0.01	−0.06–0.07	0.82	0.78–0.86	0.84	0.82–0.87
Harmful Alcohol Use	0	0–2	1.37	2.06	0.61	0.52–0.68	0	0–2	1.35	2.08	0.57	0.46–0.65	0.02	−0.06–0.09	0.88	0.85–0.91	0.94	0.93–0.95
**WHOQOL-BREF**	80	72–87	79.56	10.23	0.89	0.87–0.9	79	72–88	79.25	10.77	0.9	0.88–0.91	0.25	−0.19–0.69	0.82	0.82–0.83	0.92	0.91–0.93
Physical domain	19	17–22	19.14	3.56	0.78	0.75–0.82	19	17–22	19.16	3.64	0.8	0.76–0.83	0	−0.16–0.16	0.81	0.79–0.83	0.91	0.89–0.92
Psychological domain	21	18–23	20.51	3.19	0.8	0.76–0.83	21	18–23	20.29	3.36	0.8	0.76–0.83	0.2	0.04–0.35	0.84	0.82–0.85	0.9	0.88–0.91
Social domain	12	10–13	11.28	2.51	0.72	0.66–0.78	12	10–13	11.22	2.59	0.75	0.69–0.8	0.04	−0.05–0.13	0.84	0.81–0.87	0.94	0.93–0.95
Environmental domain	29	26–31	28.56	4.29	0.67	0.62–0.72	29	26–31	28.57	4.51	0.72	0.67–0.77	0.01	−0.18–0.19	0.83	0.81–0.84	0.92	0.91–0.94

Legend: Me—median; IQR—interquartile range; M—mean; SD—standard deviation; CI—95% confidence interval; α—Cronbach’s alpha; MoD—mean of differences (also called “bias”) calculated with Bland–Altman statistics; ICC—intraclass correlation coefficient; κ—Cohen’s kappa (unweighted).

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
