# Peer review of "The Reliability and Compatibility of the Paper and Electronic Versions of the POLLEK Cohort Study Questionnaire"

_healthcare, 2020, doi:10.3390/healthcare8040438_

Round 1

Reviewer 1 Report

I think that, despite some minor aspects, overall this is a good research.

I liked some aspects:

  1. The process of the questionnaire validation is well explained.
  2. The article is really short and it clearly explains the validation process.
  3. Overall, this is a good research with a very good methodology.

Nevertheless, I think that there are some aspects than could be improved.

  1. In the Introduction, I do not see clearly the relationship of COVID-19 with this research, as there are too many factors that can influence on the mental health problems in young medical doctors. The COVID-19 is a new factor that should almost have no influence in medical students, as they do not have to manage patients or take decisions.
  2. In the Introduction, I would suggest the authors to explain a bit better (just one or two more lines) the POLLEK questionnaire and the domains referred (AUDIT and WHOQOL-BREF).
  3. In the Methods section, I think that there is an important (and unavoidable) selection bias. First, all the students are from the first year in the Medical University of Silesia. Second, as not all of them take part in the study, there is a voluntary effect. Third, only 55.3% accomplished both versions and were included in the current analysis. This selection bias of course does not invalidate the study, but I think should be better explained. Specially, if they have considered this bias and how it has been managed or how it can affect the results.
  4. In the Discussion section, I do not see the importance of highlighting a future planned research that will be realized during the COVID-19 pandemic.
  5. In the Limitations section, I think the authors should explain the possible selection bias and how it can affect their conclusions.

But overall, these are aspects that can be revised and do not essentially affect the high quality of this research. Selection bias is important but also unavoidable. It does not invalidate the research. If it is clearly explained, it helps to better understand the conclusions.

Questionnaire validation is an essential tool nowadays. Mental health problems are important in medical doctors, and the use of online tools is actually the future (or present). Questionnaire validation process is difficult to perform and complex to explain. Therefore, I think this is a good research and a very good article. It is important to highlight that it is clear and short. The process of questionnaire validation is complex and the authors clearly and shortly reflect it.

Author Response

Dear Reviewer,

Thank you for your positive evaluation of our manuscript. Please find our point-by-point responses in attached file.

Kind regards

Reviewer 2 Report

This is an interesting paper, which needs more consistency and clarity. There are two main issues, I would like you to address in revised version.

The first package of issues is in principle related to your Introduction and objectives of your study. If I understand it correct, your paper according the title, is about validation of two different ways of administration of a survey tool, or tools as it seems to be later in text. The survey tool shall be related to your POLLOCK study; I miss a description of the POLLOCK study! Could you add more about it in Introduction? In principle, your paper is about validity issue. Yet, I am pretty sure there is a bunch of literature around survey tool validity testing methodology and you mention none of it! Again, this needs to be added, otherwise the paper is missing its objective. Third in this package, in fact you validate administration ways of two already validated survey tools. Could you please add some literature on validation of those two tools?

The second issue is clarity of results. You present many different statistical methods to assess the validity. That many, that results and becoming very hard to read and understand. Could you please improve the clarity of employed statistics and explain why and which method serves the best in individual cases?

Author Response

Dear Reviewer,

Thank you very much for the time spent on evaluating our paper and your comments. You will find our point-by-point responses to your notes in attached file.

Kind regards

Reviewer 3 Report

Since the main focus of the article is on compatibility of the paper and electronic versions of a questionnaire, the following additional information will be helpful:

Page 3, line 66: How was the paper version administered? Was it by post? Hand delivered? Or did participants have to get it from a designated point? This additional information will help to better understand why the electronic version could be attractive.

Similarly, it will be helpful to know how the online version was sent to participants. Was it emailed directly to participants as a document; as an online link; or through a software like Surveymonkey?

Author Response

Dear Reviewer,

We are grateful for your helpful comments. You will find our responses in attached file.

Kind regards

Round 2

Reviewer 2 Report

Thank you for changes and responses to my comments